# Isolation and Characterization of the Primary Marmoset (*Callithrix jacchus*) Retinal Pigment Epithelial Cells

**DOI:** 10.3390/cells12121644

**Published:** 2023-06-16

**Authors:** Ha Young Jang, Chang Sik Cho, Young Mi Shin, Jina Kwak, Young Hoon Sung, Byeong-Cheol Kang, Jeong Hun Kim

**Affiliations:** 1Fight Against Angiogenesis-Related Blindness (FARB) Laboratory, Clinical Research Institute, Seoul National University Hospital, Seoul 03082, Republic of Korea; hyjang020710@gmail.com (H.Y.J.); kansayulgi@gmail.com (C.S.C.); pgym7234@gmail.com (Y.M.S.); 2Graduate School of Translational Medicine, Seoul National University College of Medicine, Seoul 03080, Republic of Korea; jinawak@snu.ac.kr; 3Department of Experimental Animal Research, Biomedical Research Institute, Seoul National University Hospital, Seoul 03080, Republic of Korea; 4Asan Institute for Life Sciences, Asan Medical Center, Seoul 05505, Republic of Korea; 5Department of Convergence Medicine, University of Ulsan College of Medicine, Seoul 05505, Republic of Korea; 6Department of Biomedical Sciences & Ophthalmology, Seoul National University College of Medicine, Seoul 03080, Republic of Korea; 7Institute of Reproductive Medicine and Population, Seoul National University College of Medicine, Seoul 03080, Republic of Korea

**Keywords:** marmoset, retinal pigment epithelium, primary cells, age-related macular degeneration

## Abstract

Marmosets have emerged as a valuable primate model in ophthalmic research due to their similarity to the human visual system and their potential for generating transgenic models to advance the development of therapies. In this study, we isolated and cultured primary retinal pigment epithelium (RPE) cells from marmosets to investigate the mechanisms underlying RPE dysfunction in aging and age-related macular degeneration (AMD). We confirmed that our culture conditions and materials supported the formation of RPE monolayers with functional tight junctions that closely resembled the in vivo RPE. Since serum has been shown to induce epithelial–mesenchymal transition (EMT) in RPE cells, we compared the effects of fetal bovine serum (FBS) with serum-free supplements B27 on transepithelial electrical resistance (TER), cell proliferation, and morphological characteristics. Additionally, we assessed the age-related morphological changes of in vivo and primary RPE cells. Our results indicate that primary marmoset RPE cells exhibit in vivo-like characteristics, while cells obtained from an older donor show evidence of aging, including a failure to form a polarized monolayer, low TER, and delayed cell cycle. In conclusion, our primary marmoset RPE cells provide a reliable in vitro model for developing novel therapeutics for visual-threatening disorders such as AMD, which can be used before animal experiments using marmosets.

## 1. Introduction

The retinal pigment epithelium (RPE) is the pigmented cell layer, consisting of tightly interconnected regular polygonal cells that create a semi-permeable barrier between the outside choroid and inside photoreceptor cells [1]. Microvilli at the apical surface of the RPE participate in the phagocytic function, while the basolateral surface of the RPE is connected to the basal folds of Bruch’s membrane by half desmosomes. The gap, adherens, and tight junction at the apical periphery of contacting cells maintain cell polarity and control the movement of substances as the choroid–blood–retinal barrier (Figure 1) [2]. RPE function and structure alteration can lead to various retinopathies, including age-related macular degeneration (AMD), the leading cause of blindness in developed countries [3]. In AMD, the degeneration of RPE cells appears to begin with impaired phagocytosis properties, resulting in the formation of abnormal deposits called drusen, which impair the function of the RPE cells. Eventually, accumulation of drusen leads to progressive degeneration and cell death of photoreceptors and RPE, resulting in loss of central vision [4]. 

Although many animal models mimic several of the pathological features seen in AMD, they do not fully recapitulate the disease due to the unique features of the macula and complexities of AMD [5,6]. Cell culture models of RPE are valuable for understanding the RPE dysfunction and related pathophysiology in a controllable defined system. Various sources of human RPE cells are available, including primary prenatal and postnatal tissue, transformed cell lines, and stem cells, each with their extensibility, potential for differentiation, and the tendency to display RPE-like properties [7]. ARPE19 cells, a spontaneously arising human RPE-derived cell line, are not immortal and eventually exhibit replicative depletion and senescence, do not reliably express several RPE signature proteins, and do not achieve the desired differentiated phenotype [8]. Postnatal human RPE cells maintain relatively mature phenotypes in culture but have limited expansion in culture. Primary human prenatal cultures have the advantage of exhibiting many known characteristics of mature RPE and greater growth capacity than adult RPE, but they also include supply restrictions and ethical issues regarding their use [9]. Primary RPE cells have been widely utilized over the last few years to assess RPE dysfunction and basic cell biology as a more appropriate and reproducible research resource [10]. Although human cells are the most desirable for the pathophysiological study of RPE, ethical issues and the limited availability of human donors have led to studies of RPE isolation, culture, and characterization from non-human mammals [11,12,13,14]. Marmosets are a better alternative than other mammals considering the similarities with humans in ocular structure and visual function [15]. Unfortunately, to date, numerous studies have been reported on primary human RPE cells and non-human RPE cells, including those from mice and pigs, but none have been reported in marmosets [16,17,18,19,20]. 

The common marmoset (*Callithrix jacchus*), a small New World primate, provides a good model for human vision and can perform various visual identification and cognitive tasks by a human-like visual system. Marmosets mature quickly and reproduce easily in captivity, and they are suitable for genetic manipulation, which is vital in developing the first transgenic primate lines. Marmosets are ideal for genetic engineering models of ophthalmic diseases [15,21]. However, with increasing ethical considerations to reduce or replace the use of monkeys in biomedical research, the need for suitable in vitro models to complement animal models is also emerging. We aimed to utilize the primary marmoset RPE cells as a valuable in vitro model by establishing a simplified and reproducible environment for proliferation and differentiation with epithelial integrity. Moreover, we characterized age-related changes in primary marmoset RPE cells, which revealed variations depending on the donor’s age. Therefore, primary marmoset RPE cells could be an efficient screening in vitro model with scalability to cross in vivo and in vitro studies.

## 2. Materials and Methods

### 2.1. Animals 

Marmoset eyes were obtained from the Marmoset Model Network Center in Seoul National University Hospital. All marmoset experiment protocols were followed as set by the Institutional Animal Care and Use Committee of Seoul National University Hospital (IACUC No. 22-0069), an accredited research institute of the Association for Assessment and Accreditation of Laboratory Animal Care International (AAALAC) approved facility.

### 2.2. Preparation of the Growth and Maintenance Medium 

For the RPE growth medium, 5 mL of N1 supplement (Sigma-Aldrich, St. Louis, MO, USA), 5 mL of Penicillin-Streptomycin (Gibco, Waltham, MA, USA), 5 mL of nonessential amino acids (Sigma-Aldrich, St. Louis, MO, USA), 125 mg of Taurine (Sigma-Aldrich, St. Louis, MO, USA), 10 μg of Hydrocortisone (Sigma-Aldrich, St. Louis, MO, USA), 0.0065 μg of Triiodo-thyronin (Sigma-Aldrich, St. Louis, MO, USA), and 10% of heat-inactivated FBS (Gibco, Waltham, MA, USA) were added to 500 mL of DMEM. This medium can be stored at 4 °C for up to 1 month [7,18,19]. The serum-free maintenance medium contained a 7:3 mixture of high glucose DMEM (Gibco, Waltham, MA, USA) and Ham’s F12 (Gibco, Waltham, MA, USA) with 1% of Penicillin-Streptomycin. A total of 2% of 50× solution of B27 was added to the serum-free medium [7,17]. 

### 2.3. Isolation of the Primary RPE Cells

After enucleation, each eye was dipped in 70% ethanol and then placed into a 50 mL centrifuge tube containing 15 mL of medium on ice and moved within 1 h. The steps to isolate primary RPE cells were performed in a laminar flow hood under sterile conditions using sterilized instruments. The globe was placed in a 60 mm sterile culture dish after briefly soaking in 70% ethanol and washing 3 times with sterile PBS. The connective tissue and muscle attached to the sclera was carefully removed using Dumont #5 tweezers and curved iris scissors under a dissecting stereomicroscope (Nikon SMZ745T, Nikon Corporation, Tokyo, Japan). The medium was changed regularly to prevent contamination by connective tissue or blood. A step incision was performed using a sterilized #11 scalpel blade at a distance of 5 mm caudally from the limbus (Figure 2a). The anterior segment of the eye containing the corneal–iris complex was carefully removed by circumference cutting 5 mm posterior to the limbus using curved iris scissors (Figure 2b). The globe was cut into quadrants as close as possible to the optic disc for the eye cup to lay flat. It was cut enough to flatten the eyecup while keeping the petals connected (Figure 2c). The vitreous was grasped with forceps and gently pulled to remove it. A pair of tweezers was used to gently separate the retina being careful to avoid damage to the underlying RPE with the retina cut at the optic nerve attachment to facilitate separation. The neural retina was removed by gently pulling it from the edges with tweezers while holding the RPE–choroid–sclera (RCS) complex with other tweezers (Figure 2d). The RPE–choroid layer was gently peeled off from the sclera with a pair of tweezers taking care not to tear through the blunt dissection. The RPE–choroid sheet was transferred to a 60 mm sterile dish containing 3 mL of 2% (wt/vol) dispase solution (Sigma-Aldrich, St. Louis, MO, USA). It was incubated for 30 min at 37 °C in 5% CO_2_ (Figure 2e). The RPE–choroid sheet was transferred to a 60 mm sterile culture dish containing 1 mL of medium following washing the sheet 3 times using the pre-warmed medium to stop the dispase activity. RPE cells were harvested by repeated gentle aspiration from the RPE–choroid sheet with a P200 micropipette while avoiding choroidal aspiration. The RPE cells were washed 3 times with the pre-warmed medium. After each washing step, the cells were carefully collected, avoiding other tissues, such as choroid debris. The RPE cells were pelleted by centrifuge at 340× *g* for 2 min at RT. The supernatant was discarded. The cells were carefully resuspended in 1 mL of medium. The cells were then placed into a T75 flask with an additional 10 mL of the growth medium volume and incubated at 37 °C, in 5% CO_2_. The cells were cultured for 72 h without the change of medium. After 3 days, the medium was changed twice per week. After 1 week the cells became approximately more than 90% confluent. 

### 2.4. Culture of Polarized Monolayer RPE Cells

The Transwell membranes with 0.4 μm pores (Corning Costar, Cambridge, MA, USA) were prepared by coating the upper compartment membrane with fibronectin (5 μg/mL^−1^) dissolved in 100 μL of ddH_2_O for 2 h. The fibronectin was then removed by aspiration and the membrane rinsed with PBS. The coated membranes were air-dried overnight at RT in the laminar flow hood [18]. After the first week of isolation, the medium was removed, and the cells in the T75 flask were washed with 10 mL of PBS. The PBS was then removed and 2 mL of 1× EDTA-trypsin was added. Trypsinization was usually complete within 10 min. The cells were resuspended in medium to stop the trypsin reaction, centrifuged at 340× *g* for 2 min at RT, and the supernatant was discarded. The cells were gently pipetted with the serum-free medium and the RPE suspension plated at a density of 1 × 10^5^ cells per well (200 μL) drop-wise into the center of the prepared Transwell inserts. The cells were checked to ensure they were evenly distributed under the microscope. A volume of 700 μL of the serum-free medium was introduced into the lower compartment. The Transwells were placed in an incubator at 37 °C, 5% CO_2_. The medium was completely changed every 3–4 days. Within up to 6 weeks, polarized RPE monolayers were characterized by measuring TER and immunohistochemistry. In addition, to assess whether the composition of the medium was associated with the generation of polarized RPE monolayers, we cultured cells under the same condition in medium containing 10% FBS. 

### 2.5. Transepithelial Resistance (TER) Measurement

TER measurements were taken using an EVOM3 epithelial volt-ohmmeter and STX4 electrode (World Precision Instruments, Sarasota, FL, USA) according to the manufacturer’s instruction. The electrodes were sterilized with 70% ethanol, rinsed in ddH_2_O, and equilibrated in a pre-warmed culture medium. Measurement was performed in the hood within 10 min after the cells were taken out from the incubator. Net TERs (Ω cm^2^) were calculated by subtracting the value of a blank insert, fibronectin-coated membrane without cells, from the experimental value and multiplying it by the area of the insert membrane.

### 2.6. Immunohistochemistry of RPE Cells 

The cells grown on the Transwell insert membrane were washed twice with PBS and fixed with 4% paraformaldehyde for 15 min at 37 °C. A volume of 500 μL was added to the top and 1000 μL to the bottom chamber. After PBS washing, the membranes were removed from the inserts with a sterile #11 scalpel blade. The cells were then permeabilized with 0.25% Triton X-100 for 5 min and blocked with 1% BSA in PBS for 30 min at RT. Antibodies were diluted in 1% BSA/PBS and incubated with the cells overnight at 4 °C (as described in Appendix A). After washing with PBS, the cells were incubated for 1 h at RT in the dark with the Alexa Fluor-conjugated secondary antibody (Invitrogen, Carlsbad, CA, USA); in this case the Alexa Fluor-conjugated primary antibody was not used. The cells were washed with PBS and counterstained with 1 μg/mL 4′,6′-diamidino-2-phenylindole (DAPI) for 5 min. They were placed on a glass slide with the cell side up, covered with a coverslip, and fluorescence images were obtained using a laser scanning confocal microscope (Leica TCS STED, Leica Microsystem Ltd., Wetzlar, Germany). We analyzed four cellular morphometric parameters, including cell area, perimeter, aspect ratio, and circularity, from images obtained via confocal microscopy, according to the Image J user guidelines. We excluded the cells at the image edges and those that could not be fully identified. The spatial scale of the images was defined so that the measurement result could be displayed in calibrated units in μm. We calculated the area (μm^2^) and length (μm) of the selected region by selecting the outside boundary of the f-actin or ZO-1-stained each cell. Aspect ratio and circularity were calculated using the shape descriptor of Image J in the selected cell area. Aspect ratio (major axis/minor axis) is calculated as the ratio between the major and minor axes of an ellipse, and this parameter is higher in elongated cells. Circularity ([4π(area/perimeter)^2^]) is a shape descriptor that mathematically indicates the degree of similarity to a perfect circle, with a value close to 1.0 designating a perfect circle and a value close to 0.0 indicating a less circular shape.

### 2.7. Immunohistochemistry of RCS Complex

RCS complexes were carefully separated from the contralateral eye of the donor and fixed with 4% paraformaldehyde using a dissecting microscope (Nikon SM2745T, Nikon Corporation, Tokyo, Japan). The RCS complexes were permeabilized by incubating them with Perm/Block solution (0.2% Triton-X 100 and 0.3% BSA in PBS) for 2 h at RT. They were then incubated with antibodies diluted with Perm/Block solution (0.2% Triton-X 100 and 0.3% BSA in PBS) overnight at 4 °C, using the antibodies described in Appendix A. The RCS complexes were washed with PBS and counterstained with 1 μg/mL 4′,6′-diamidino-2-phenylindole (DAPI) for 15 min. After washing with PBS, the RCS complexes were placed on a glass microscope slide with the RPE layer facing up. After coverslip, fluorescence images were obtained using a laser scanning confocal microscope (Leica TCS STED, Leica Microsystem Ltd., Wetzlar, Germany). Based on the research finding that the location within the eye can affect cell growth, we classified the RPE whole mount into 3 regions: central, equatorial, and peripheral. Six images were randomly acquired from each region, and the cell morphology was analyzed using the same method [22,23].

### 2.8. Crystal Violet Staining Assay

RPE cells were seeded at 1 × 10^4^ cells per wells of a 12-well plate with medium containing 10%, 5%, and 2.5% FBS or in the serum-free medium. After 3, 5, or 7 days of culture, each well was washed with PBS and fixed with 4% paraformaldehyde. Cells were stained with 0.5% crystal violet solution (Sigma-Aldrich, St. Louis, MO, USA) for 30 min. The plate was washed 5 times with PBS and air-dried. Crystal violet was eluted from the cells by incubation with 1% SDS solution for at least 1 h and the absorbance was read at 570 nm under a microplate reader (Infinite^®^ 200 pro, Tecan, Switzerland). ARPE19 cells cultured in medium containing 10% FBS were used as a control group for quantification of primary marmoset RPE cells. 

### 2.9. Cell Cycle Analysis

RPE cells were harvested and fixed with 70% ethanol for 1 h at 4 °C. The cells were washed twice with PBS, and stained with Propidium iodide/RNAses solution (Invitrogen, Carlsbad, CA, USA) for 30 min before being examined using a flow cytometer (Accuri C6 Plus, BD Biosciences, Franklin Lakes, NJ, USA). The data were analyzed using a BD Accuri C6 software (Version 1.0.264.21, BD Biosciences, Franklin Lakes, NJ, USA). 

### 2.10. Western Blot Analysis

The protein extracts were prepared using RIPA buffer (1% Triton X-100). The protein contents were measured using the BCA protein assay kit (Pierce, Rockford, IL, USA). Proteins were separated using SDS-PAGE and transferred onto polyvinylidene fluoride (PVDF) membranes (Merck Millipore Ltd., Tullagreen, Ireland). The membrane was incubated in primary antibody diluted with 1% non-fat milk overnight at 4 °C with gentle rocking. The membrane was washed 3 times and incubated with the appropriate diluted secondary antibody for 1 h at RT. The antibodies used for immunoblotting are summarized in Appendix A. Immunoreactive bands were visualized using ImageQuant^TM^ LAS4000 (GE Healthcare, Chicago, IL, USA). 

### 2.11. Statistical Analysis

The results are presented as the mean ± SD. Differences between two groups were assessed with the two-tailed Student’s unpaired *t*-test. The one-way ANOVA with Tukey’s multiple comparison test was used to assess differences between more than two groups as indicated in figure legends. Statistical analysis was performed with GraphPad Prism (San Diego, CA, USA) version 7.0 software. Data were considered statically significant at *p* < 0.05 (* *p* < 0.05, ** *p* < 0.01). 

## 3. Results

### 3.1. Characterization of RPE Cells

After 24 h of initial incubation using the RPE medium containing 10% FBS, cells of a marmoset aged 6 days old were attached to T75 flasks with high pigmentation (Figure 3a). The cells with high pigmentation proliferated rapidly (Figure 3b), exhibiting packed round shape colonies (Figure 3c). After 7 days, the typical morphology of RPE monolayer resembling RPE in vivo was visible (Figure 3d). Proteins abundantly expressed in the RPE, including RPE65, were identified through Western blot (Figure 3e), similar to ARPE19 cells (Figure 3f).

The cultured monolayer’s similarity to natural RPE was assessed with TER measurement and immunohistochemistry. In Figure 4a, it is shown that RPE cells were able to adhere and form highly differentiated monolayers with a high TER for over six weeks using B27 alone. Notably, a high TER of 163.65 ± 4.16 Ω cm^2^ was achieved after only 2 weeks of Transwell culture, which significantly increased to 229.65 ± 7.27 Ω cm^2^ at 6 weeks. The generated RPE monolayer showed regular polygonal geometrical features with ZO-1 and f-actin overlapping in apical-lateral contact with neighboring cells, similar to those observed in the contralateral eye of the same donor (Figure 4b). 

### 3.2. Changes in Cellular Morphology Depending on the Culture Medium

To assess the effect of FBS on the formation of monolayers with functional tight junctions during the initial RPE culture, the cells were cultured in a medium containing 10% FBS or a serum-free medium for 14 days after reaching confluency. Although the cells achieved a high TER of 181.19 ± 10.99 Ω cm^2^ in the 10% FBS-containing medium, the tight junction function was significantly impaired compared to the serum-free medium (Figure 5a). The presence of FBS in the medium resulted in an overall reduction in the area of cell-to-cell contact, and it also induced morphological changes such as an increase in cell size and elongated spindle-like shapes that resemble those observed during an epithelial–mesenchymal transition (EMT) (Figure 5b,c). We reduced the time required to form polarized monolayers with a high TER (204.99 ± 8.64 Ω cm^2^) to 14 days by changing to the serum-free medium after cells reached confluency. In addition, it was suggested that the serum-free medium is more suitable for forming highly differentiated monolayers with high TER than the FBS-containing medium in primary marmoset RPE cell culture. 

### 3.3. Changes in Cellular Morphology and TER according to Donor Age 

Primary RPE cells from a 7-year-old and a 6-day-old marmoset were isolated and cultured in Transwell inserts for six weeks using the serum-free medium. The RPE cells isolated from the 7-year-old marmoset exhibited significantly low TER (100.69 ± 2.58 Ω cm^2^) than cells isolated from the 6-day-old marmoset (214.29 ± 4.42 Ω cm^2^) (Figure 6a). In RPE cells isolated from the 6-day-old marmoset, we observed that ZO-1 and F-actin were highly expressed at the periphery of the cells and colocalized with each other, forming a clear line at the lateral boundaries between adjacent cells. The polygonal cells formed a regular geometric pattern by contacting their neighbor cells, facilitating morphological analysis (Figure 6c). The cells had an area of 279.25 ± 27.77 μm^2^, a perimeter of 65.16 ± 3.54 μm, an aspect ratio of 1.47 ± 0.29, and a circularity of 0.83 ± 0.05 on average. The cells derived from the 7-year-old marmoset exhibited fragmented ZO-1 expression at the cell borders, making it difficult to identify individual cells for morphological analysis. F-actin was distributed not only along the cell border but also inside the cell with stress fibers and hardly colocalized with ZO-1 (Figure 6b). The cells grew into multiple layers without contact with adjacent cells after they reached confluency. The morphological analysis could not be carried out because identifying the boundaries of individual cells was challenging. 

To assess whether isolated RPE cells formed a similar structure to in vivo RPE, the RCS complex of the opposite eye of each donor was stained with ZO-1 and f-actin for immunohistochemistry. The RPE was observed in three geographic locations of the RPE flat mounts (Figure 7a). In the central and equatorial RPE of the 7-year-old marmoset, ZO-1 and f-actin were not only distributed along the cell border but also scattered throughout the cell inside. At the cell border, ZO-1 was fragmented or not colocalized with f-actin, which made it challenging to identify the boundaries of individual cells. In the peripheral region, ZO-1 and f-actin were mainly localized and colocalized at cell borders, enabling the identification of cell borders (Figure 7b). In the 6-day-old marmoset, ZO-1 and F-actin were highly expressed at the cell border and colocalized throughout the eyeball, making it easier to identify the cell boundaries. The cells formed a regular geometric pattern with hexagonal shapes (Figure 7c). While aspect ratio and circularity showed no significant differences among the regions, cell area and perimeter were significantly higher in the central region than in the equatorial and peripheral regions in the RPE of the 6-day-old marmoset (Figure 7d). The cell area and perimeter of the polarized monolayer of primary RPE cells were similar to the RPE in the equatorial region. However, it may depend on the seeding cell density. 

### 3.4. Changes in Cell Proliferation with Donor Age and Growth Medium Composition

To evaluate the effects of donor age and fetal bovine serum (FBS) concentration in medium on the proliferation of primary RPE cells, we cultured cells isolated from a 7-year-old and a 6-day-old marmoset monkeys in media containing 0%, 2.5%, 5%, or 10% FBS for 7 days. We then stained the cells with crystal violet to assess proliferation (Figure 8a). The proliferation of RPE cells isolated from the 7-year-old marmoset in growth media containing 10% FBS significantly decreased compared to cells isolated from the 6-day-old marmoset. As a reference with ARPE cells, the most commonly used cells in RPE research, cell proliferation of cells isolated from the 6-day-old marmoset was 15.78% higher, while the cells isolated from the 7-year-old marmoset were around 12.64% lower in media containing 10% FBS (Figure 8b). The proliferation of RPE cells isolated from the 7-year-old (Figure 8c) and the 6-day-old marmoset (Figure 8d) significantly decreased as the concentration of FBS in the growth media decreased. Unlike ARPE19 cells, primary RPE cells did not show significant proliferation in the serum-free medium regardless of donor age. This suggests that culture medium may differ depending on the cellular proliferation and maturation of primary RPE cells as an in vitro model, emphasizing the importance of selecting appropriate media based on the experimental objectives. 

### 3.5. Changes in Cell Cycle and Proteins Involved in Cell Cycle with Donor Age 

To investigate whether the decrease in proliferative potential and morphological changes in RPE cells isolated from a 7-year-old marmoset was due to cellular senescence, we analyzed cell cycle progression. RPE cells isolated from the 7-year-old marmoset exhibited an increased distribution in the G0/G1 phase, while the percentage of cells in the S phase decreased compared to cells isolated from the 6-day-old marmoset (Figure 9a,b). Additionally, the protein levels of cyclin D1, cyclin E, and CDK6 were significantly reduced, whereas p53 and p21 were markedly upregulated in the cells isolated from the 7-year-old marmoset (Figure 9c,d). These results indicate that RPE cells isolated from the 7-year-old marmoset exhibited G0/G1 arrest. 

## 4. Discussion

We aimed to provide the potential of primary marmoset RPE cells as an in vitro model by developing a simple and defined method for their isolation, proliferation, and differentiation while maintaining epithelial integrity, with the characterization of cells from donors of different ages. There are several cell culture models to study RPE biology, including spontaneously formed cell lines (e.g., ARPE19), immortalized cell lines (e.g., hTERT-RPE1, RPE-J, and D407), primary human or animal RPE cells, and embryonic and induced pluripotent stem cell (iPSC) derived RPE [24]. ARPE-19 cells are widely used due to their ability to mimic many in vivo hallmarks when cultured under conditions that promote differentiation [25]. However, they have abnormal karyotypes and have been passaged over many years, reducing the ability to demonstrate the hallmarks of differentiated RPE. Additionally, cell phenotypes such as hexagonicity, pigmentation, junction resistance, and polarity are considered medium to low compared to primary RPE cells [18]. Primary RPE cultures were established from freshly harvested RPE obtained from mouse, pig, or human donors [24]. The time to maturity varied by species, but cultures formed well-differentiated monolayers with tight junctions and high TER. Unfortunately, while there has been a report of differentiating RPE from marmoset embryonic stem cells, no primary cultures of RPE have been reported In marmosets [16,26]. The common marmoset (*Callithrix jacchus*) possesses typical ocular features as a primate, and genetic modification is relatively more feasible to generate than other laboratory monkeys, making them valuable for research on congenital visual disorders [15]. The advantages of the in vitro culture models are potential reliability, consistency, and low cost compared to in vivo studies. Development and characterization of primary marmoset RPE cells as an in vitro model that recapitulates the RPE’s morphology and function in vivo can be a reliable high-throughput platform for preclinical studies before using marmosets. 

The cell culture model of RPE in marmosets can play an important role in better understanding RPE cell biology and RPE dysfunction related to AMD pathogenesis. Primary RPE cultures have been established from mouse, porcine, and human donors [16]. The use of marmosets to obtain primary RPE cells is not more widely accessible than rodents and is not supported by the reference provided, which details characterization of isolated primary mouse RPE [19]. Since many cells could be isolated from one eye, various molecular biological analyses, including proteomic analysis, were possible without cell pooling. In addition, primary mouse RPE could not maintain high TER, pigmentation, and similar morphology in vivo after division with trypsin [19]. In contrast, marmoset cells maintained morphological characteristics and high TER after trypsin division and cryopreservation. Most of the commercial antibodies in this study responded well to antigen–antibody reactions, even though there was no application information in monkeys. RPE cells isolated from newborn marmoset monkeys within one week of birth exhibit similar characteristics to those of humans, including high pigmentation, a polarized monolayer with high TER, and RPE-specific expression proteins, including ZO-1, E cadherin, ezrin, f-actin, and RPE65. These cells attach to fibronectin-coated Transwell membranes in the serum-free media and, over 6 weeks, achieve high TER. After reaching confluency in serum-containing media, they form similar geometric patterns of RPE within 2 weeks in the serum-free medium. ZO-1 and f-actin were mainly observed at the periphery of the cells and co-localized, facilitating the demarcation of cell boundaries, allowing morphological analysis, including cell area (μm^2^), perimeter (μm), aspect ratio, and circularity of each cell. The morphological features of matured primary RPE cells were similar to the RPE in vivo from the same donor. Morphological analysis revealed that the cells exhibited a comparable cell area and perimeter to the equatorial region of the RPE whole mount. However, it may vary depending on the cell seeding density. As is known, FBS in the media inhibits the formation of monolayers in primary marmoset RPE cells and induces morphological changes, such as increased cell size and an EMT-like spindle shape. Although serum is necessary for the growth and initial adhesion of RPE cells in culture plates, its presence in culture medium can be problematic due to various unknown hormones, cytokines, and growth factors. Exposure to conventional serum levels retracts experimental reproducibility due to a variable and batch-dependent mixing of growth factors, resulting in morphological heterogeneity, dedifferentiation, and EMT-like phenotypes [27,28]. However, there was a notable difference in the proliferative potential of RPE cells between the serum-free medium with B27 supplements and media containing FBS. B27 supplement, a highly enhanced form of N2 initially designed to support the maintenance of hippocampal neurons and neuronal cell lines, can support the proliferation and maintenance of RPE cells even in the absence of N2 supplement [7,29,30]. Similarly, in the serum-free media supplemented with B27, cells primarily attached and reached confluency naturally over time, followed by adopting a compact, polygonal morphology and repigmentation over six weeks. The addition of serum shortened the time required to reach confluency, and subsequently the serum-free media supported the formation of a polarized monolayer with regular geometric features and high TER. Despite serum exposure during sample collection persisting as serum contaminants despite multiple washings and incubation in the serum-free medium [31], the characteristic morphological integrity of the RPE cells was exhibited after at least 14 days of the serum-free culture. TER measurement is commonly used to measure the establishment of functional tight junctions between RPE cells with high sensitivity and reliability. Primary marmoset RPE cells, which formed monolayers with functional tight junction using the serum-free media, achieved a high TER of over 200. It is higher than the low TER (<50 Ω cm^2^) observed in cell lines such as ARPE19 and D407, and comparable to the typical TER value for human RPE monolayer in vivo (approximately 150 Ω cm^2^) and cultured RPE monolayer (25–200 Ω cm^2^) [18,32,33,34]. 

Primary RPE cells undergo dedifferentiation by losing their pigmentation and characteristic morphology and acquire a fibroblastic morphology during passages, which is considered useful for studying the epigenetic assessment of AMD progression [24]. The primary RPE cells in humans are ideal for research, but there are limitations, such as the availability of donor tissue, restrictions on the age of available donors, and the need to collect cells within a short post-mortem interval. Marmoset RPE culture has several advantages, including high visual and immune systems similarity with humans and a relatively feasible use of donor tissues with a wide range of donor ages. In addition, it was possible to compare single-cell and tissue levels by observing in vivo changes in the contralateral eye of the donor. Morphological changes were exhibited in the cells isolated from a 7-year-old marmoset, including increased cell size, loss of uniform geometry, and inability to overlap ZO-1 and f-actin at the cellular boundary. These morphological changes were also observed similarly in vivo, particularly prominent in the central and equatorial regions of the RPE whole mount. Isolated RPE cells also did not form a polarized monolayer with high TER. RPE cells form a monolayer network through cell–cell adhesion, including tight junctions with neighboring cells. Tight junctions are a type of cell–cell adhesion that functions as a paracellular by blocking the movement of plasma and toxic substances into the retina and controlling the flow of fluids and ions across the RPE from the choroid to the retina. ZO-1 is a membrane-associated tight junction adapter protein connecting transmembrane proteins to the cytoplasmic actin for initial formation and distinct organization of tight junctions [2]. Depletion of ZO-1 in MDCK cells results in the specific defects in the barrier for large solutes, accompanied by morphological changes and reorganization of apical actin and myosin [35]. Overexpression of ZONAB of knockdown of its cellular inhibitor ZO-1 leads to enhanced proliferation of RPE cells and induced structural changes an EMT-like phenotype [36]. The presence of rich stress fibers in the f-actin cytoskeleton was directly correlated with poor phagocytic activity in RPESC-RPE cells, while the lateral circumferential f-actin and the lack of stress fibers were directly associated with a high phagocytic activity [37]. RPE cells isolated from a 7-year-old marmoset exhibited an abundance of stress fibers and a lack of contiguous lateral circumferential f-actin, indicating a potential decline in RPE cell function, including phagocytic activity. The improper colocalization of ZO-1 and f-actin leads to compromised cell-to-cell junctions with neighboring cells, potentially resulting in decreased TER and impaired barrier function. Along with morphological changes, signs of cellular senescence, such as decreased cell proliferation and delayed G0/G1 phase, were identified, suggesting that RPE cells isolated from aged marmosets could be an in vitro model for age-related RPE dysfunction, including AMD. 

We isolated RPE cells from marmosets for the first time and cultured native RPE monolayers that closely resembled those observed in the donor’s RPE in vivo. We facilitated the formation of highly differentiated RPE monolayers with high TER by using B27 supplements instead of serum. We also aimed to characterize primary RPE cells isolated from a 7-year-old marmoset to identify the morphological changes that occur naturally in marmosets. The cells exhibited known age-related morphological changes, including cell size increase, uniform geometric loss, abundant stress fibers, and loss of ZO-1 and f-actin colocalization at cell boundaries with signs of cellular senescence. While further studies are required to determine whether primary marmoset RPE cells recapitulate the morphology and function of the RPE, primary marmoset RPE cultures can serve as an in vitro model to develop effective novel therapies in the future. 

## 5. Conclusions

Primary marmoset RPE cells share similar features with RPE cells in vivo and primary human RPE cells, making them suitable for molecular studies. In addition, RPE cells isolated from the older marmoset exhibit age-related changes. Establishing an in vitro model based on the RPE of marmosets can promote the identification of epigenetic factors underlying RPE dysfunction, including AMD, and enable the development of cell-based model systems that mimic the morphological and functional abnormalities associated with AMD. This model provides a reliable and high-throughput platform for preclinical testing of potential therapies and pharmaceuticals, which can be combined with animal experiments using marmosets. 

## Figures and Tables

**Figure 1 cells-12-01644-f001:**
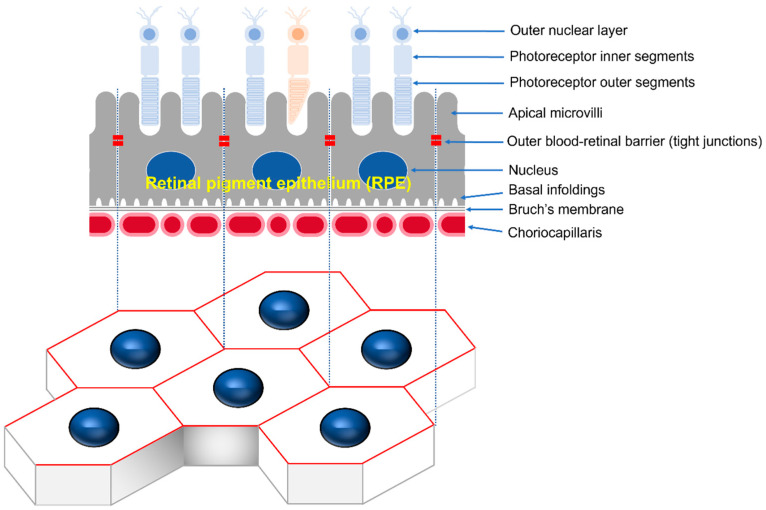
Schematic drawing of the RPE monolayer. RPE refers to a single layer of regular, polygonal RPE cells arranged at the outermost layer of the neural retina. The outside of the RPE is connected to Bruch’s membrane and the choroid, while the inside is connected to the outside of photoreceptor cells. The tight junctions formed between the upper lateral membranes of RPE cells are responsible for a permeability barrier known as the outer blood–retinal barrier.

**Figure 2 cells-12-01644-f002:**
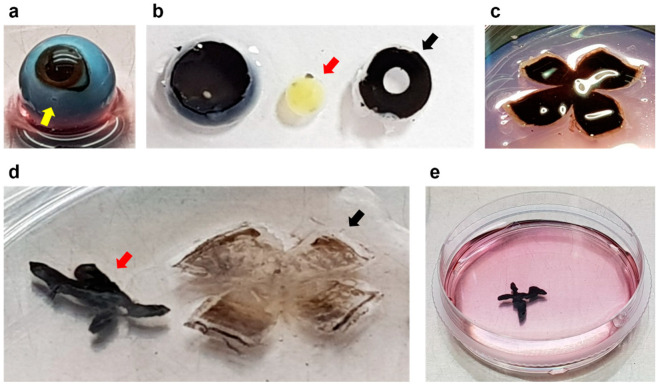
Dissection of the marmoset eyes. Dissection processing for the isolation of the primary RPE cells in marmoset. (**a**), Marmoset eye with the first step incision at a distance of 5 mm caudally from the limbus (yellow arrow). (**b**), The eye split into cornea–iris (black arrow), lens (red arrow), and posterior eye cup. The anterior segment is removed at once and is easily separated into the lens and cornea–iris complex. (**c**), The posterior eye cup is cut into quadrants to lay flat. (**d**) The retina (black arrow) and RPE–choroid sheet (red arrow) were gently separated. (**e**) RPE–choroid sheet incubating with 2% (wt/vol) dispase solution for 30 min at 37 °C in 5% CO_2_.

**Figure 3 cells-12-01644-f003:**
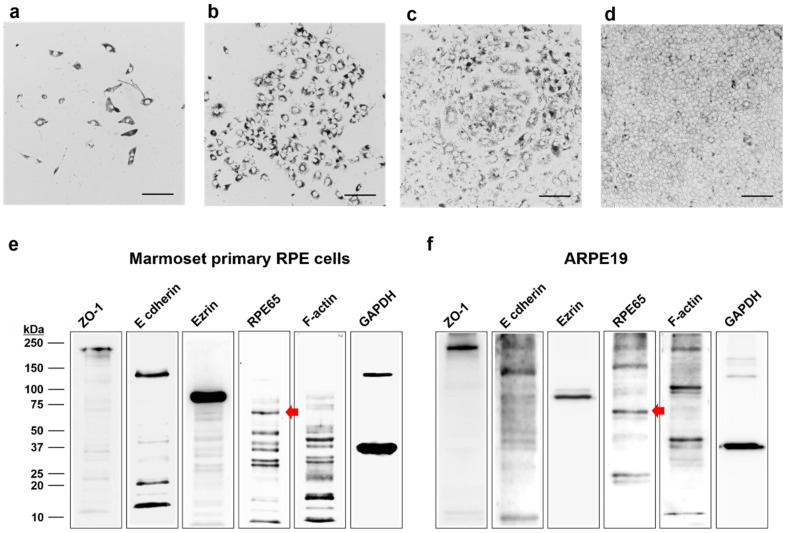
Light microscopic images and Western blot analysis of primary marmoset RPE cells. Light microscopic images show the time-dependent growth pattern of RPE cells in a T75 flask. (**a**) Day 1 cells were attached, and the initial growth of the culture with high pigmentation. (**b**) Day 3 cells were grown rapidly while maintaining high pigmentation. (**c**) Day 5 cells continued to undergo rapid proliferation while forming round-shaped colonies. (**d**) Day 7 cells in an early passage reached confluency and appeared with a regular polygonal shape on a T75 flask. (**e**,**f**) Western blot analysis of ZO-1 (molecular weight: 220 kDa), E-cadherin (molecular weight: 135 kDa), ezrin (molecular weight: 81 kDa), f-actin (molecular weight: 42 kDa), GAPDH (molecular weight: 37 kDa), and RPE65 (molecular weight: 61 kDa, red arrow), the RPE-specific protein, in the primary marmoset RPE cells (**e**) and ARPE19 cells (**f**). Scale bar, 20 μm.

**Figure 4 cells-12-01644-f004:**
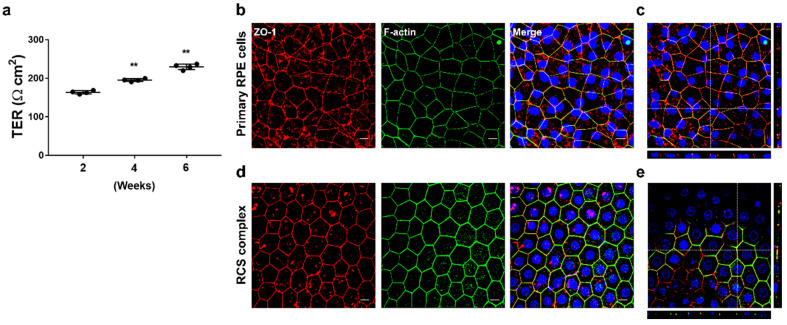
Characterization of RPE monolayers in comparison to the RPE in vivo. Monolayers with functional tight junctions were generated by culturing for 6 weeks in the serum-free medium alone in a fibronectin-coated Transwell insert and comparing the morphological features of RPE in the contralateral eye of the same donor. (**a**) Time-dependent increase in TER from multiple Transwells containing primary marmoset RPE cells from a single donor. (**b**) Merged images of primary marmoset RPE cells for immunofluorescence patterns for tight junction protein ZO-1 and f-actin visualized by confocal microscopy. (**c**) Upper panel: En-face view of a polarized RPE layer through the *z*-axis. Bottom and right panel: Cross-section through the *z*-plane of multiple optical slices at the location indicated by the white reference line in the corresponding upper panel. (**d**) Merged images of RCS complex for immunofluorescence in vivo patterns for ZO-1 and f-actin. (**e**) Upper panel: En-face view of the RCS complex through the *z*-axis. Bottom and right panel: Cross-section through the *z*-plane of multiple optical slices at the location indicated by the white reference line in the corresponding upper panel. Scale bar, 10 μm. Data are presented as the mean ± SD and analyzed by ANOVA. Tukey post hoc test: ** *p* < 0.01.

**Figure 5 cells-12-01644-f005:**
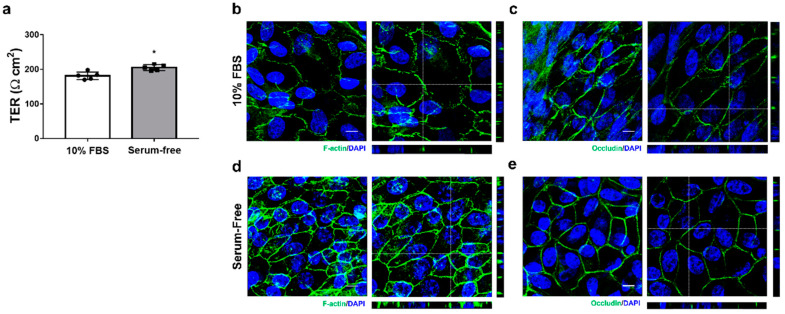
Comparison of RPE monolayer formation ability. The generation of RPE monolayers in a fibronectin-coated Transwell insert was compared between a serum-free medium and a medium containing 10% FBS. (**a**) TER increased with culture in the serum-free medium. (**b**,**c**) Immunofluorescence patterns for ZO-1 and occludin visualized by confocal microscopy in the RPE cells cultured with the growth medium containing 10% FBS for 13 days. Cell size increased and morphological features changed to an elongated-spindle shape. (**d**,**e**) Immunofluorescence patterns for ZO-1 and occludin visualized by confocal microscopy in the RPE cells cultured for 10 days with the serum-free medium after 3 days incubation with the growth medium containing 10% FBS. Scale bar, 10 μm. Data are presented as the mean ± SD and analyzed by two-tailed Student’s unpaired *t*-test: * *p* < 0.05.

**Figure 6 cells-12-01644-f006:**
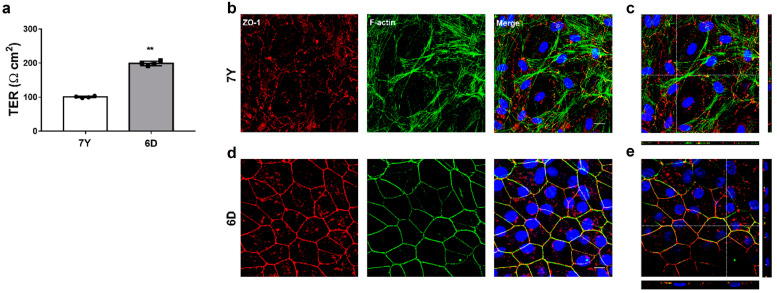
Comparison of morphological characteristics of primary RPE cells depending on donor’s age. RPE cells isolated from a 7-year-old and a 6-day-old marmoset were cultured for 6 weeks in the serum-free medium alone in a fibronectin-coated Transwell insert. (**a**) TER significantly decreased in RPE cells isolated from the 7-year-old marmoset. (**b**) Immunofluorescence patterns for tight junction protein ZO-1 and f-actin visualized by confocal microscopy in RPE cells isolated from the 7-year-old marmoset. RPE cells failed to form monolayers with functional tight junctions in the features of overlapping ZO-1 and f-actin at the periphery of each cell. (**c**) Upper panel: En-face view of a polarized RPE layer through the *z*-axis. Bottom and right panel: Cross-section through the *z*-plane of multiple optical slices at the location indicated by the white reference line in the corresponding upper panel. (**d**) Immunofluorescence patterns for tight junction protein ZO-1 and f-actin visualized by confocal microscopy in RPE cells isolated from the 6-day-old marmoset. (**e**) Upper panel: En-face view of the RCS complex through the *z*-axis. Bottom and right panel: Cross-section through the *z*-plane of multiple optical slices at the location indicated by the white reference line in the corresponding upper panel. Scale bar, 10 μm. Data are presented as the mean ± SD and analyzed by two-tailed Student’s unpaired *t*-test: ** *p* < 0.01.

**Figure 7 cells-12-01644-f007:**
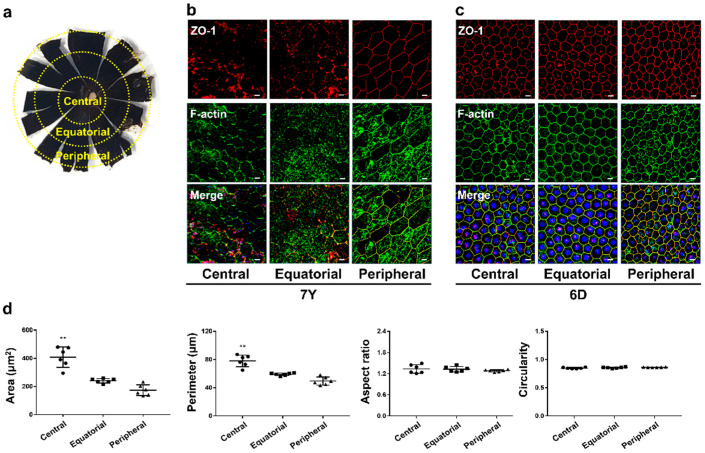
Characterization of age-related morphological changes in marmoset RPE. (**a**) RPE flat mount image showing the different geographic locations from which the images were obtained. (**b**) Immunofluorescence patterns for tight junction protein ZO-1 and f-actin visualized by confocal microscopy in RCS complex of the 7-year-old marmoset. Compared to the 6-day-old marmoset, morphological features are changed, including increased cell size, loss of uniform geometry, and inability to overlap ZO-1 and f-actin at the cell interface. (**c**) Immunofluorescence patterns for tight junction protein ZO-1 and f-actin visualized by confocal microscopy in RCS complex of the 6-day-old marmoset. (**d**) Morphological analysis of the 6-day-old marmoset was performed on six images, each from categorized locations within the eye, including cell area (μm^2^), perimeter (μm), aspect ratio, and circularity. Scale bar, 10 μm. Data are presented as the mean ± SD and analyzed by ANOVA. Tukey post hoc test: ** *p* < 0.01.

**Figure 8 cells-12-01644-f008:**
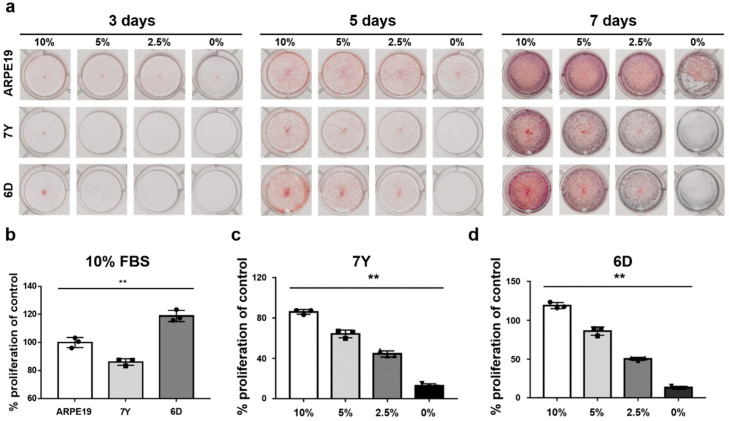
Proliferative potential of primary RPE cells changed by donor’s age and FBS concentration in culture medium. Crystal violet staining was used to assess the proliferative changes of RPE cells isolated from a 7-year-old and a 6-day-old marmoset and ARPE19 cells depending on the various concentration of FBS in medium. ARPE19 cells cultured in medium containing 10% FBS were used as a control group for quantification of primary marmoset RPE cells. (**a**) Cells in 12-well plates stained with 0.5% crystal violet after culture for 3, 5, and 7 days in medium containing 10%, 5%, 2.5% FBS, and the serum-free medium. (**b**) RPE cells isolated from the 7-year-old marmoset significantly decrease proliferation even in 10% FBS-containing medium as measured by microplate reader by eluting crystal violet. (**c**,**d**) RPE proliferation significantly decreases as the FBS content decreases in the medium regardless of the donor’s age. Data are represented as the mean ± SD and analyzed by ANOVA. Tukey post hoc test: ** *p* < 0.015.

**Figure 9 cells-12-01644-f009:**
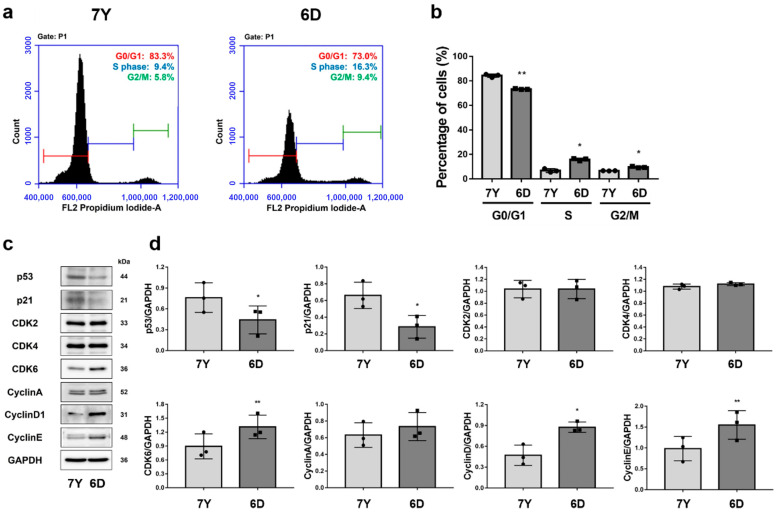
RPE cells isolated from a 7-year-old marmoset showed cell cycle delay in G0/G1 phase. Cell cycle analysis and Western blot assay for cell cycle protein expression perform to assess whether the cell proliferation was associated with an age-related cell cycle arrest in primary marmoset RPE cells. (**a**,**b**) In cell cycle analysis, RPE cells from the 7-year-old marmoset (7Y) are in cell cycle delay at G0/G1 and decrease in percentages of S and G2/M phases compared to RPE cells from the 6-day-old marmoset (6D). (**c**,**d**) RPE cells isolated from the 7-year-old marmosets show decreased cyclin D and cyclin E while elevated p53 and p21 expression levels, compared to RPE cells from the 6-day-old marmoset, suggesting the suppression of G1-to-S-phase progression. Data are represented as the mean ± SD and analyzed by ANOVA. Tukey post hoc test: ** *p* < 0.01 and * *p* < 0.05.

## Data Availability

The data that support the findings of this study are available from the corresponding author upon reasonable request.

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
