# Peer review of "Isolation and Characterization of the Primary Marmoset (Callithrix jacchus) Retinal Pigment Epithelial Cells"

_cells, 2023, doi:10.3390/cells12121644_

Round 1
Reviewer 1 Report
The manuscript entitled “Isolation and Characterisation of the Primary Marmoset (Callithrix Jacchus) Retinal Pigment Epithelial Cells” by Jang HY. et al. reviews the generation of primary marmoset RPE cells as a new model for preclinical study in retinal diseases. The manuscript is well written and clear.
I have several comments to improve the manuscript:
1. Introduction: there is no mention of a previous article (Torrez LD, et al. 2012) in which is demonstrated the derivation of RPE cells from marmoset pluripotent stem cells.
2. Methods section: the number of eyeballs from 6 and 7-year-old marmoset are used in this study.
3. Results: In the “immunohistochemistry of RPE cells”, the author should add more RPE markers as MITF, PAX-6, Tyrp-1, Ezrin or BEST.
4. The authors could quantify the VEGF secretion or the phagocytic activity of marmoset RPE cells, in order to highlight the cell functionality.
5. An important parameter in the generation of a cell line is the cryopreservation. To this aim, the authors should evaluate the cell survival after a freezing and thawing cycles.

Author Response
We sincerely appreciate your feedback and the reviewers' careful consideration and insightful comments. Your comments and suggestions are of great importance in improving the quality of our research. We have thoroughly reviewed the reviewer's comments and have utilized the "Track Changes" feature to revise the manuscript accordingly. Additionally, we have prepared a cover letter containing detailed point-by-point responses.
We would be honored to receive continued guidance and expertise from both the editors and the reviewers as we strive to refine the manuscript further. Your valuable input will undoubtedly contribute to the overall enhancement of our work.
Reviewer 1
Comments and Suggestions for Authors
The manuscript entitled “Isolation and Characterisation of the Primary Marmoset (Callithrix Jacchus) Retinal Pigment Epithelial Cells” by Jang HY. et al. reviews the generation of primary marmoset RPE cells as a new model for preclinical study in retinal diseases. The manuscript is well written and clear.
I have several comments to improve the manuscript:
- Introduction: there is no mention of a previous article (Torrez LD, et al. 2012) in which is demonstrated the derivation of RPE cells from marmoset pluripotent stem cells.
: While the previous study by Torrez LD et al. focused on differentiating RPE cells from marmoset embryonic stem cells, it is important to note that our study utilized primary RPE cell culture with a distinct cell source and isolation method. However, we appreciate the reviewer's feedback and have considered it. In response, we have provided additional information in the Discussion section (line 446) and referenced citation 26 (https://doi.org/10.1155/2012/417865), which reports the culture of marmoset RPE cells derived from pluripotent stem cells predating our primary RPE cell culture. This reference helps highlight the existing literature and the different approaches to studying marmoset RPE cells.
- Methods section: the number of eyeballs from 6 and 7-year-old marmoset are used in this study.
: A 6-day-old and a 7-year-old marmoset were used, and this information was added in the figure legend (6-9) and results.
- Results: In the “immunohistochemistry of RPE cells”, the author should add more RPE markers as MITF, PAX-6, Tyrp-1, Ezrin or BEST.
: RPE65, a specific 65kDa protein in the retinal pigment epithelium (RPE), is a crucial marker for RPE due to its essential role in maintaining the visual cycle in mammalian RPE. In numerous studies on the isolation and cultivation of primary RPE cells, the expression of RPE65, either alone or in conjunction with other markers such as ZO-1, has been used to validate the RPE cells (https://doi.org/10.3791/56997)(https://doi.org/10.1038/nprot.2009.33). While markers like MITF and PAX6 are also important in RPE and associated with melanogenesis. BEST is known as a channel protein that regulates chloride ion flow across the basolateral membrane of RPE cells.
Our purpose was to assess whether isolated RPE cells from marmosets exhibit regular geometric patterns similar to those observed in vivo, while establishing functional tight junctions. To achieve this, we conducted western blotting of the expression of ZO-1 (a tight junction protein), ezrin (a membrane-organizing protein), e-cadherin (a Ca2+-dependent cell adhesion molecule), and f-actin (a key cytoskeletal component). Additionally, in Figure 3, we employed western blotting to validate the affinity of commercially available antibodies for marmosets and to confirm the RPE cells.
- The authors could quantify the VEGF secretion or the phagocytic activity of marmoset RPE cells, in order to highlight the cell functionality.
: We aimed to assess whether marmoset primary RPE cells differentiate into characteristic morphologies accompanied by functional tight junctions. Since f-actin morphology is known to be associated with phagocytic activity (https://doi.org/10.1016/j.stemcr.2018.01.017), we emphasized the observation of morphology rather than cellular function. Additionally, we intended to focus on the potential of primary RPE cells to recapitulate in vivo by forming similar morphology to those observed in vivo from the same donor.
- An important parameter in the generation of a cell line is the cryopreservation. To this aim, the authors should evaluate the cell survival after a freezing and thawing cycles.
: We isolated and characterized primary RPE cells from marmosets, which did not undergo additional tumorization or immortalization. Similar to human or mouse primary RPE cells, we observed evidence of cellular senescence, such as decreased cell division with repeated passages.

Reviewer 2 Report
Overall this a good paper with methods described in sufficient detail to allow replication and uptake of the method by other research groups. The authors have primarily focused on the proliferative abilities and morphology of cells isolated from different ages of marmoset and under medium containing different serum concentrations providing a good indication as to how differing culture conditions may be used to achieve a particular experimental objective. In this respect the marmoset RPE have been characterised well and the publication is informative in terms of cell culture set up. However little work has been performed to convince the reader that uptake of marmoset RPE is preferential to other methods of RPE cell culture including the use of ARPE-19 cells. This is required from both an ethical and cost perspective when utilising marmoset RPE. The work is extremely promising and is the first of its kind but the reviewer would like to see more evidence to support retention of an RPE phenotype before the manuscript is accepted for publication. This would confirm the credibility of marmoset RPE as a novel RPE source for in vitro culture and would promote more widespread uptake by the field. A few examples of additional work that should to be included are as follows:
A) demonstrating polarisation by either immunofluorescence staining (Na+/K+ ATPase or MerTK) and/or SEM images showing membrane specialisation and the polarised distribution of organelles. B) Characterising cytokine secretion profiles in the media (VEGF/PEDF) which could be easy to obtained from identical experimental set ups in combination with ELISA. C. qPCR to demonstrate increased expression of key RPE markers relative to commonly used cell lines (ARPE-19 cells) D. Quantification of pigmentation/pigment granules. The images in Figure 3a-d are certainly supportive of this however it is difficult to comment on when no comparator group has been included. E. Demonstration that primary marmoset RPE recapitulate RPE function e.g. photoreceptor internalisation assays and to what degree.
Please see specific comments on the manuscript in relation to each section below. Where typos/grammatical errors have been picked up these have been highlighted along with suggestions. The reviewer hopes that these are helpful in polishing the final version of this promising manuscript.
Abstract: The introductory sentence is a little confusing. Do the authors mean that the isolation of RPE cells from transgenic primate models has the potential to bridge the gap between in vivo and in vitro studies. If so this needs to be made clearer. At present it reads that the use of transgenic primate models themselves bridge the gap which is considered in vivo work. Perhaps the authors could also allude to the fact that this as a refinement to and reduction in animal use under the 3Rs principles in the abstract. Similarly, the use of serum-free B27 supplements would also be considered a reduction under the 3Rs.
Line 3: Callithrix Jacchus > Callithrix jacchus.
Line 41: Brush’s > Bruch’s
Line 56: as the outer blood-retinal barrier > known as the outer blood-retinal barrier
Line 58: macular> macula
Line 116-118: Under a dissecting stereomicroscope (Nikon SMZ745T, NY, USA), carefully removed the connective tissue and muscle attached to the sclera using Dumont #5 tweezers and curved iris scissors. > The connective tissue and muscle attached to the sclera was carefully removed using Dumont #5 tweezers and curved iris scissors under a dissecting stereomicroscope (Nikon SMZ745T, NY, USA).
Line 126-127: Using a pair of tweezers, the retina gently separated the retina while avoiding damage to the RPE and cut it at the optic nerve attachment. > A pair of tweezers was used to gently separate the retina being careful to avoid damage to the underlying RPE with the retina cut at the optic nerve attachment to facilitate separation.
Line 135: wash> washing
Line 154: Could the authors please state the membrane pore size and substrate of the Transwell membranes used in the study for the purposes of reproducibility as membrane pore size has been shown to significantly influence the polarisation and differentiation properties of RPE in cell culture. It would also be useful to include how many Transwell plates of particular well sizes can be set up from a single marmoset eye. To this end a point could be made in the discussion as to how this could achieve a reduction/refinement in animal use.
Line 163-165: The cells were gently pipetted with the serum-free medium and placed the RPE suspension at a density of 1 × 105 cells per well (200 μl) drop-wise to the center of the prepared Transwell inserts. > The cells were gently pipetted with the serum-free medium and the RPE suspension plated at a density of 1 × 105 cells per well (200 μl) drop-wise into the centre of the prepared Transwell inserts.
Line 168: Please state whether this was a complete media change or alternatively the percentage of media changed every 3-4 days.
Line 173: resistnace > resistance
Line 193: place> placed
Line 213: incubate>incubated
Line 216: The cells proliferated rapidly with high pigmentation (Figure 3b), packed round 261 shape colony (Figure 3c).> The cells with high pigmentation proliferated rapidly (Figure 3b), exhibiting packed round shape colonies (Figure 3c)
Figure 3: Has any quantification been performed as to the relative expression levels of proteins in isolated primary marmoset RPE vs. ARPE-19 cells to demonstrate that this is a superior model. This is required to demonstrate the relevance of isolating RPE by this method for cell culture studies on RPE function/disease processes from both a scientific and ethical perspective. The reviewer requests that this be included in the manuscript. From the western blots presented it also appears that ZO-1 and RPE-65 expression are decreased in primary marmoset RPE. Quantification is needed to clarify this. The two red arrows also appear to be above the 61KDa band for RPE-65, please amend this.
Figure 4: Please show individual antibody stains for ZO-1 and F-actin for both primary marmoset RPE and contralateral eyes in addition to merged images shown for ease of interpretation. It would also be useful to show improved barrier function by another quantifiable method e.g. FITC-Dextran diffusion.
Line 336: fivers> fibers
Line 369: mages>images
Figure 7: How many individual eyes were analysed for quantification purposes. Do data points correspond to numbers obtained from n=6 RCS. Please make this clear in the figure legend.
Line 396: uses > was used
Line 400: stain > stained
Line 404: Is standard deviation presented here, the methods section states that SM is presented for all experimental results. Please keep this consistent.
Line 413: Remove ‘cells suggest that RPE’.
Line 418: perform >perform
Discussion: The discussion is good and highlights the limitations of the ARPE-19 cell line and benefits of primary RPE sources. Similarly, it discusses the benefits of utilising marmoset RPE and states the opportunity to develop and characterise primary marmoset RPE as an in vitro model that recapitulated RPE morphology and function yet limited work has been done to this effect in the results section.
Line 452: Please rephrase. The use of marmosets to obtain primary RPE is not more widely accessible than rodents and is not supported by the reference provided which details characterisation of isolated primary mouse RPE.
Line 462: This contradicts the previous statement (line 456) ‘primary mouse RPE could not maintain high TER’. The authors also emphasise the fact that cells exhibit polarisation however little work has been performed to support this statement.
Line 500: dedifferentiate >dedifferentiation
Please see comments and suggestions for authors section on typos and grammatical points picked up during the review process. The reviewer hopes these are useful in polishing the final version of the manuscript. I would however suggest a thorough proof check prior to publication.
Author Response
We sincerely appreciate your feedback and the reviewers' careful consideration and insightful comments. Your comments and suggestions are of great importance in improving the quality of our research. We have thoroughly reviewed the reviewer's comments and have utilized the "Track Changes" feature to revise the manuscript accordingly. Additionally, we have prepared a cover letter containing detailed point-by-point responses.
We would be honored to receive continued guidance and expertise from both the editors and the reviewers as we strive to refine the manuscript further. Your valuable input will undoubtedly contribute to the overall enhancement of our work.
Reviewer 2
Overall this a good paper with methods described in sufficient detail to allow replication and uptake of the method by other research groups. The authors have primarily focused on the proliferative abilities and morphology of cells isolated from different ages of marmoset and under medium containing different serum concentrations providing a good indication as to how differing culture conditions may be used to achieve a particular experimental objective. In this respect the marmoset RPE have been characterised well and the publication is informative in terms of cell culture set up. However little work has been performed to convince the reader that uptake of marmoset RPE is preferential to other methods of RPE cell culture including the use of ARPE-19 cells. This is required from both an ethical and cost perspective when utilising marmoset RPE. The work is extremely promising and is the first of its kind but the reviewer would like to see more evidence to support retention of an RPE phenotype before the manuscript is accepted for publication. This would confirm the credibility of marmoset RPE as a novel RPE source for in vitro culture and would promote more widespread uptake by the field. A few examples of additional work that should to be included are as follows:
- A) demonstrating polarisation by either immunofluorescence staining (Na+/K+ATPase or MerTK) and/or SEM images showing membrane specialisation and the polarised distribution of organelles. B) Characterising cytokine secretion profiles in the media (VEGF/PEDF) which could be easy to obtained from identical experimental set ups in combination with ELISA. C. qPCR to demonstrate increased expression of key RPE markers relative to commonly used cell lines (ARPE-19 cells) D. Quantification of pigmentation/pigment granules. The images in Figure 3a-d are certainly supportive of this however it is difficult to comment on when no comparator group has been included. E. Demonstration that primary marmoset RPE recapitulate RPE function e.g. photoreceptor internalisation assays and to what degree.
Please see specific comments on the manuscript in relation to each section below. Where typos/grammatical errors have been picked up these have been highlighted along with suggestions. The reviewer hopes that these are helpful in polishing the final version of this promising manuscript.
- Abstract: The introductory sentence is a little confusing. Do the authors mean that the isolation of RPE cells from transgenic primate models has the potential to bridge the gap between in vivoand in vitro If so this needs to be made clearer. At present it reads that the use of transgenic primate models themselves bridge the gap which is considered in vivo work. Perhaps the authors could also allude to the fact that this as a refinement to and reduction in animal use under the 3Rs principles in the abstract. Similarly, the use of serum-free B27 supplements would also be considered a reduction under the 3Rs.
: The first sentence of the Abstract has been rewritten to clarify the value of marmosets in ophthalmological research.
- Line 3: Callithrix Jacchus > Callithrix jacchus.
: We have addressed the reviewer's comments and made the necessary revisions accordingly.
- Line 41: Brush’s > Bruch’s
: We corrected a typographical error.
- Line 56: as the outer blood-retinal barrier > known as the outer blood-retinal barrier
: We have addressed the reviewer's comments and made the necessary revisions accordingly.
- Line 58: macular> macula
: We corrected a typographical error.
- Line 116-118: Under a dissecting stereomicroscope (Nikon SMZ745T, NY, USA), carefully removed the connective tissue and muscle attached to the sclera using Dumont #5 tweezers and curved iris scissors. > The connective tissue and muscle attached to the sclera was carefully removed using Dumont #5 tweezers and curved iris scissors under a dissecting stereomicroscope (Nikon SMZ745T, NY, USA).
: We have addressed the reviewer's comments and made the necessary revisions accordingly.
- Line 126-127: Using a pair of tweezers, the retina gently separated the retina while avoiding damage to the RPE and cut it at the optic nerve attachment. > A pair of tweezers was used to gently separate the retina being careful to avoid damage to the underlying RPE with the retina cut at the optic nerve attachment to facilitate separation.
: We have addressed the reviewer's comments and made the necessary revisions accordingly.
- Line 135: wash> washing
: We have addressed the reviewer's comments and made the necessary revisions accordingly.
- Line 154: Could the authors please state the membrane pore size and substrate of the Transwell membranes used in the study for the purposes of reproducibility as membrane pore size has been shown to significantly influence the polarisation and differentiation properties of RPE in cell culture. It would also be useful to include how many Transwell plates of particular well sizes can be set up from a single marmoset eye. To this end a point could be made in the discussion as to how this could achieve a reduction/refinement in animal use.
: The transmembrane used in our study has a pore size of 0.4 μm, stated in line 157.
- Line 163-165: The cells were gently pipetted with the serum-free medium and placed the RPE suspension at a density of 1 × 105cells per well (200 μl) drop-wise to the center of the prepared Transwell inserts. > The cells were gently pipetted with the serum-free medium and the RPE suspension plated at a density of 1 × 105 cells per well (200 μl) drop-wise into the centre of the prepared Transwell inserts.
: We have addressed the reviewer's comments and made the necessary revisions accordingly.
- Line 168: Please state whether this was a complete media change or alternatively the percentage of media changed every 3-4 days.
: Medium was completely changes, as indicated in line 171.
- Line 173: resistnace > resistance
: We corrected a typographical error.
- Line 193: place> placed
: We corrected a typographical error.
- Line 213: incubate>incubated
: We corrected a typographical error.
- Line 216: The cells proliferated rapidly with high pigmentation (Figure 3b), packed round 261 shape colony (Figure 3c).> The cells with high pigmentation proliferated rapidly (Figure 3b), exhibiting packed round shape colonies (Figure 3c)
: We have addressed the reviewer's comments and made the necessary revisions accordingly, as in line 261.
- Figure 3: Has any quantification been performed as to the relative expression levels of proteins in isolated primary marmoset RPE vs. ARPE-19 cells to demonstrate that this is a superior model. This is required to demonstrate the relevance of isolating RPE by this method for cell culture studies on RPE function/disease processes from both a scientific and ethical perspective. The reviewer requests that this be included in the manuscript. From the western blots presented it also appears that ZO-1 and RPE-65 expression are decreased in primary marmoset RPE. Quantification is needed to clarify this. The two red arrows also appear to be above the 61KDa band for RPE-65, please amend this.
: We did not perform a quantitative analysis of protein levels in Figures 3e and 3f. The purpose of western blotting in Figure 3 was twofold: (1) to validate the affinity of commercially available antibodies for use in marmoset monkeys and (2) to confirm the isolated RPE cells. As these experiments served as preliminary investigations preceding the studies presented in Figures 4-9, we are willing to consider the inclusion of Figures 3e and 3f as supplementary data. Additionally, if the editor grants approximately 4 additional weeks, we could conduct additional experiments for quantification.
However, there are several questions that we agree that a quantitative comparison of ZO-1 or RPE65 protein expression in ARPE19 cells and marmoset RPE cells is suitable for demonstrating a superior model. ARPE19 cells are a spontaneous cell line derived from humans, (1) the possibility of species differences with marmoset monkeys, and (2) the antibody used for western blot was not verified in monkeys according to the datasheet provided by the manufacturer. Consequently, we aimed to assess the differentiation of single cells into monolayers that resemble those observed in vivo while establishing a high TER.
- Figure 4: Please show individual antibody stains for ZO-1 and F-actin for both primary marmoset RPE and contralateral eyes in addition to merged images shown for ease of interpretation. It would also be useful to show improved barrier function by another quantifiable method e.g. FITC-Dextran diffusion.
: In Figure 4, we have included separated images stained with ZO-1 and f-actin. We appreciate the reviewer for suggesting a method to evaluate barrier function, such as FITC-dextran diffusion. In our laboratory, perfusion experiments have often been employed to assess barrier function in mouse experiments. However, in marmoset monkeys, we typically minimize the number of animals used by utilizing various organs obtained during necropsy for multiple experiments. Consequently, when FITC-dextran is systemically administered, there is a possibility of interference in various experiments involving organs other than the eyes. These limitations have prompted us to initiate studies using cells isolated from marmosets. We are currently continuing our research with marmoset-derived cells, and we plan to conduct various evaluations and studies, including the barrier function assessment suggested by the reviewer.
- Line 336: fivers> fibers
: We corrected a typographical error.
- Line 369: mages>images
: We corrected a typographical error.
- Figure 7: How many individual eyes were analysed for quantification purposes. Do data points correspond to numbers obtained from n=6 RCS. Please make this clear in the figure legend.
: The data in Figure 7 were quantified by performing whole-mount staining on eyes obtained from a 6-day-old marmoset monkey and a 7-year-old marmoset monkey. The intraocular location was categorized into regions according to Figure 7a, and 6 images were acquired and analyzed within each region in a 6-day-old marmoset. This information is described in the figure legend.
- Line 396: uses > was used
: We corrected a typographical error.
- Line 400: stain > stained
: We corrected a typographical error.
- Line 404: Is standard deviation presented here, the methods section states that SM is presented for all experimental results. Please keep this consistent.
: We have addressed the reviewer's comments and made the necessary revisions accordingly. SM in the Statistics section of Materials and Methods needed to be corrected. The statistical analysis we conducted is represented by the standard deviation (SD). This information has been consistently revised and clarified in each figure legend.
- Line 413: Remove ‘cells suggest that RPE’.
: We have addressed the reviewer's comments and made the necessary revisions accordingly.
- Line 418: perform >perform
: We corrected a typographical error.
- Discussion: The discussion is good and highlights the limitations of the ARPE-19 cell line and benefits of primary RPE sources. Similarly, it discusses the benefits of utilising marmoset RPE and states the opportunity to develop and characterise primary marmoset RPE as an in vitromodel that recapitulated RPE morphology and function yet limited work has been done to this effect in the results section.
: In line 552 of the Discussion section, we emphasized the need for further research on the morphology and barrier function of the marmoset primary RPE.
- Line 452: Please rephrase. The use of marmosets to obtain primary RPE is not more widely accessible than rodents and is not supported by the reference provided which details characterisation of isolated primary mouse RPE.
: We have addressed the reviewer's comments and made the necessary revisions accordingly.
- Line 462: This contradicts the previous statement (line 456) ‘primary mouse RPE could not maintain high TER’. The authors also emphasise the fact that cells exhibit polarisation however little work has been performed to support this statement.
: We have removed the mention of mice in line 462. The retinal pigment epithelium (RPE) exhibits polarity with distinct apical and basolateral plasma membrane domains separated by tight junctions (https://doi.org/10.1016/j.ceb.2019.08.001). Therefore, the localization of functional tight junctions and membrane transporters is closely associated with cell polarity (RPE Polarity and Barrier Function | SpringerLink). TER is a widely employed quantitative technique for evaluating tight junction integrity by measuring the electrical resistance across a cellular monolayer. Previous studies characterizing RPE-derived cells have reported high TER as an indication of functional tight junctions and polarization of the RPE (https://doi.org/10.1038/nprot.2009.33). In our study, we observed the apico-lateral ZO-1 and high TER in marmoset primary RPE cells, which we used as indicators of polarization. However, based on the reviewer's feedback, we have revised the manuscript to use more functional tight junctions rather than the term "polarization."
- Line 500: dedifferentiate >dedifferentiation
: We corrected a typographical error.

Round 2
Reviewer 1 Report
I find that you have not answered n5, in which I spoke about cryopreservation/thawing, and not concerrning the cellular passage.
See below
An important parameter in the generation of a cell line is the cryopreservation. To this aim, the authors should 50 evaluate the cell survival after a freezing and thawing cycles. 51 : We isolated and characterized primary RPE cells from marmosets, which did not undergo additional tumorization or 52 immortalization. Similar to human or mouse primary RPE cells, we observed evidence of cellular senescence, such as 53 decreased cell division with repeated passages.
Author Response
Comments and Suggestions for Authors
I find that you have not answered n5, in which I spoke about cryopreservation/thawing, and not concerrning the cellular passage.
See below
An important parameter in the generation of a cell line is the cryopreservation. To this aim, the authors should 50 evaluate the cell survival after a freezing and thawing cycles. 51 : We isolated and characterized primary RPE cells from marmosets, which did not undergo additional tumorization or 52 immortalization. Similar to human or mouse primary RPE cells, we observed evidence of cellular senescence, such as 53 decreased cell division with repeated passages.
: We agree with the reviewer’s comment that evaluating cell proliferation and viability after cryopreservation/thaw cycles is crucial for establishing cell lines. However, in our study, we specifically investigated primary marmoset RPE cells, which were not immortalized cell line. We believe that more evidence is required to establish primary RPE cells from freshly harvested marmoset tissue as a spontaneously arising cell line. In addition, in our study, we aimed to focus on the phenotypes, including morphological characteristics and functional tight junctions similar to the cells in vivo. In primary mouse RPE cells, high TER and pigmentation have been reported to be maintained only at passage 0 (https://www.doi.org/10.1038/nprot.2016.065). In addition, ARPE19, the most widely used RPE cell line, also has limitations such as low TER and morphological changes (https://www.doi.org/10.1016/j.exer.2022.109046). Therefore, primary RPE cultures were attempted to propose a cell culture model that can complement existing cell lines. However, there was evidence of cellular senescence in primary marmoset RPE cells with passages and cryopreservation/thaw cycles. So, we are trying to generate marmoset RPE cell line through gene editing. Comments and suggestions from reviewer will greatly improve the quality of our next study on immortalized marmoset RPE cell line.

Reviewer 2 Report
The reviewer thanks the authors for incorporating the suggested alterations. I would however strongly encourage you to conduct advanced characterisation of the cells (polarisation/function etc) in an additional manuscript to maintain the association and novelty of marmoset RPE with your research group.
Author Response
The reviewer thanks the authors for incorporating the suggested alterations. I would however strongly encourage you to conduct advanced characterisation of the cells (polarisation/function etc) in an additional manuscript to maintain the association and novelty of marmoset RPE with your research group.
: We greatly appreciate your valuable feedback. This manuscript represents the beginning of our research to expand the potential of marmosets in biomedical research, and we continue to conduct bioresource-related studies involving marmosets. In addition, the primary marmoset RPE cells are currently being utilized in the process of creating an immortalized cell line through gene editing techniques. Incorporating the reviewer’s comments will help us achieve more meaningful results in our following research using marmoset RPE cells.
